# Studies on the Formation of Catalytically Active PGM Nanoparticles from Model Solutions as a Basis for the Recycling of Spent Catalysts

**DOI:** 10.3390/molecules27020390

**Published:** 2022-01-08

**Authors:** Martyna Rzelewska-Piekut, Zuzanna Wiecka, Magdalena Regel-Rosocka

**Affiliations:** Institute of Chemical Technology and Engineering, Poznan University of Technology, ul. Berdychowo 4, 60-965 Poznań, Poland; zuzanna.g.wiecka@doctorate.put.poznan.pl (Z.W.); magdalena.regel-rosocka@put.poznan.pl (M.R.-R.)

**Keywords:** platinum group metals (PGM), platinum(IV), palladium(II), rhodium(III), ruthenium(III), PGM nanoparticles

## Abstract

The paper presents basic studies on the precipitation of platinum, palladium, rhodium, and ruthenium nanoparticles from model acidic solutions using sodium borohydride, ascorbic acid, and sodium formate as reducing agents and polyvinylpyrrolidone as a stabilizing agent. The size of the obtained PGM particles after precipitation with NaBH_4_ solution does not exceed 55 nm. NaBH_4_ is an efficient reducer; the precipitation yields for Pt, Pd, Ru, Rh are 75, 90, 65 and 85%, respectively. By precipitation with ascorbic acid, it is possible to efficiently separate Pt, Rh, and Ru from Pd from the two-component mixtures. The obtained Pt, Pd, and Rh precipitates have the catalytic ability of the catalytic reaction of p-nitrophenol to p-aminophenol. The morphological characteristic of the PGM precipitates was analyzed by AFM, SEM-EDS, and TEM.

## 1. Introduction

PGMs (*platinum group metals*) are among the most important in the metal industry. The main advantages of PGMs include the ability to catalyze various chemical reactions, chemical and mechanical resistance, and resistance to high temperatures [1]. As a result, the demand for these metals is growing, while their natural sources are increasingly limited. An alternative may be the recovery of metals from secondary resources, including spent chemical and petrochemical catalysts [2,3].

PGMs form the active layer of automotive converters. Platinum and/or palladium are responsible for the oxidation of hydrocarbons and CO, while NO_x_ reduction is carried out with rhodium [4]. In search of economic catalytically active metals, research on replacing the expensive Rh (price in October 2021: 13,800 $/Oz) with a cheaper Ru (price in October 2021: 659 $/Oz) has been carried out [5] due to catalysts containing Ru can be used to reduce NO_x_ [6,7,8]. Therefore, in the present studies, except for Pt, Pd and Rh, also Ru was included.

As the synthesis of PGM nanoparticles (NPs) is likely to have an important role in the future for catalytic reactions because nanoparticles are more active than catalysts of a large size [9], the authors have decided to focus on NP formation from leach solutions from secondary resources. The use of smaller than micrometric particles is advantageous and can improve the catalytic properties of the material and decrease the consumption of PGM due to the extended active surface. PGM NPs can be used as catalysts in various reactions: alcohol oxidation, CO oxidation, ethylene glycol reforming (Pt-NPs) [10,11,12], the Suzuki reaction, hydrogenation of alkene (Pd-NPs) [13,14], Fischer–Tropsch reaction (Ru-NPs) [15] and hydrogenation of benzene and phenylacetylene (Rh-NPs) [16]. Metal nanoparticles (NPs) have found applications as catalysts because of their increased selectivity and activity compared to those of traditional construction. This is due to the high surface-to-volume ratios, which provide a large number of active sites per unit of area compared to their mass counterparts [17,18].

Due to the use of significant amounts of PGM in industry and thus the high demand for these metals, new and efficient methods of recovering these metals are investigated. For this purpose, hydrometallurgical methods can be applied to recover PGMs from converters to leach solutions and to recycle PGMs by synthesis of catalytically active nanomaterials. This approach, i.e., metal recovery and reuse, is a key concept in converting waste into resources (WTR) in a circular economy.

NPs (small materials that range in size from 1 to 100 nm) are of great interest to scientists because they have unique properties compared to their bulk counterparts. Depending on the synthesis, the materials may differ in morphology, structure and size [19]. NPs can be synthesized in a variety of ways and fall into two main classes, i.e., top-down and bottom-up approaches. Synthesis by the top-down approach consists of physically reducing particles of greater mass into smaller particles of nanometric size [20]. Several researchers have proposed mechanical milling [21] and laser ablation [22] for this purpose. The nanostructures are also formed on the substrate using a bottom-up synthesis method, which involves stacking atoms on top of each other to form crystal planes, which are then stacked on top of each other to form nanostructures [18,19]. Chemical vapor deposition (CVD) [23] wet synthesis [24], sol-gel method [25], and self-assembly processes [26] are among the technique proposed in the bottom-up approach. Chemical reduction is often used to obtain a precipitate of a certain size and particle morphology [27,28,29]. The main reducing agents used for the precipitation of PGM are hydrazine [30,31], sodium borohydride [32,33,34], ascorbic acid [35,36,37] and formic acid [38,39]. On the one hand, the advantages of chemical reduction include the simplicity and rate of NPs synthesis. On the other hand, the disadvantages of this method of NP formation are the toxicity and high price of reducing agents [19].

The scientific goal of the research is to form NPs of PGM (Pt, Pd, Rh, Ru) from one- and two-component model solutions as a base for the recycling of spent automotive converters. The influence of the precipitation parameters on the size, structure, morphology and catalytic properties of the obtained particles/nanoparticles is determined. The effects of the type and concentration of a reducer, addition of a stabilizing agent, and pH on the size and morphology of the particles are also investigated. The development of an effective reduction method from leach solutions may have a significant impact on the recycling of precious metals from spent automotive converters and reuse of PGM, leading to limitation of metal mining from natural resources. Thus, the proposed fundamental research from various model solutions is vital to establish conditions that could be applied for the real leach solutions (e.g., after leaching of various waste materials-spent automotive converters) to produce catalytically active materials that may have a high application potential in the future, e.g., obtaining energy in the form of hydrogen through photoreforming from biomass or wastewater treatment from organic substances.

## 2. Materials and Methods

### 2.1. Reagents and Solutions

One-component model solutions were prepared by dissolving in 0.1 M HCl the required amounts of PtCl_4_ (96%), PdCl_2_ (99.9%), RhCl_3_ (99.9%) and RuCl_3_ (99.9%) supplied by Sigma Aldrich (Schnelldorf, Germany). Polyvinylpyrrolidone PVP (M_w_ ≈ 55000, Sigma Aldrich) was used as the stabilizing agent. Sodium borohydride NaBH_4_ (>98.0%, Sigma Aldrich), ascorbic acid C_6_H_8_O_6_ (AA, p.a., Chempur, Piekary Śląskie, Poland), sodium formate HCOONa, and formic acid HCOOH (p.a., Sigma Aldrich) were used as reducing agents in the study.

To obtain NPs, scientists proposed the use of PVP as a stabilizing agent. Depending on the synthesis, it can be used as a surface stabilizer, growth modifier, and dispersant for NPs. As a stabilizer, PVP prevents particle aggregation caused by repulsive forces. This is because the polymer contains hydrophobic carbon chains that extend into the solvents and interact with each other as a steric hindrance effect [40,41].

### 2.2. Synthesis of Nanoparticles

The stabilizing agent PVP was added to the appropriate amount of the PGM precursor and mixed for 10 min. After proper mixing, the reducer was added drop by drop, at the mixing speed of 500 rpm. The pH of the solution was then adjusted to neutral/basic (pH 7–8) with 1 M Na_2_CO_3_. The molar ratio of the metal precursor to the reducing agent and to the stabilizing agent was 1:(1 or 2):(1 or 5) depending on the experiment. The precipitation yield (P) was calculated according to the formula:(1)$$P=\frac{m_{0}-m_{p}}{m_{0}}\times 100\%$$
where m_0_ is the initial mass of the metal ions in the solution before precipitation and m_p_ is the mass of the metal ions in the solution after precipitation. The mass change of the metal ions in the solution before and after precipitation was calculated on the basis of the AAS technique (Atomic Absorption Spectrometry) determination of the metal ion concentrations. The samples after precipitation were centrifuged before AAS analysis (time: 15 min, 15,700 rpm).

### 2.3. Catalytic Reaction

The prepared precipitates were subjected to the catalytic reaction of p-nitrophenol (NPh) to p-aminophenol (APh) in the presence of NaBH_4_ as a hydrogen ion donor source following the reduction reaction:(2)$$4{\;\mathrm{NO}}_{2}C_{6}H_{6}O^{-}+3{\;\mathrm{BH}}_{4}^{-}\;\overset{\mathrm{PGM}}{\to }\;4{\;\mathrm{NH}}_{2}C_{6}H_{6}O^{-}+3{\;\mathrm{BO}}_{2}^{-}+2{\;H}_{2}O$$

The reduction of NPh to APh by sodium borohydride is a thermodynamically favorable reaction (E_0_ for 4-nitrophenol/4-aminophenol −0.76 V) [42]. The mechanism includes adsorption of both reactants (NPh and BH_4_^−^ ions) on the surface of PGM-NPs before NPh reduction, followed by the production of hydrogen radicals by electron transfer. Hydrogen species are added to adsorbed NPhs to form APh through the formation of the 4-hydroxylaminophenol intermediate and the removal of two water molecules from the nitro group. Finally, the product–APh–is detached from the surface of PGM-NP to provide a place for another catalytic cycle [43,44].

NaOH was added to the resulting solution to raise pH to 11.5 and keep NPh in anionic form. The UV-Vis spectrum of the basic form of NPh has a maximum at a wavelength of 400 nm. The spectra were recorded 25 min after the initiation of the reaction to monitor changes in NPh concentration, i.e., the addition of the reducing agent and the appropriate amount of PGM-NP catalyst. The conversion degree (α_NPh_) was calculated from Equation (3):(3)$$\alpha_{\mathrm{NPh}}=\frac{C_{0,\mathrm{NPh}}-C_{p,\mathrm{NPh}}}{C_{0,\mathrm{NPh}}}\times 100\%$$
where C_0,NPh_ is the concentration of NPh before the reaction and C_p,NPh_ is the concentration of NPh after the reaction.

### 2.4. Apparatus

The atomic absorption spectrometer (ContrAA 300, Analytik Jena, Jena, Germany) was used for the measurement of metal ion concentrations in aqueous samples at the following wavelengths: 266.0, 244.8, 343.5, 349.9, nm for Pt(IV), Pd(II), Rh(III), Ru(III) respectively. SEM-EDS (SEM FEI Quanta 250 FEG), digital microscope of the VHX-7000 series (Keyence International, Mechelen, Belgium), Hitachi HT7700 transmission electron microscope (Hitachi, Tokyo, Japan) working in high contrast and high-resolution mode, and AFM (NX10, Park Systems, Mannheim, Germany) were applied to analyze the structure and morphology of nanoparticles obtained of precipitation stages from model solutions. The solutions after reduction were analyzed by UV-Vis (spectrophotometer, Specord 40, Analytik Jena, Jena, Germany) to confirm the presence of nanoparticles.

## 3. Results and Discussion

### 3.1. Effect of PVP Concentration on PGM-NP Formation

The results of the production of palladium particles without a stabilizing agent from a one-component solution (2 mM Pd(II) in 0.1 M HCl) confirmed a positive impact of formic acid (4 mM HCOOH) as a palladium reducing agent. The volume ratio of the precursor and the precipitant was 1:1. The tests were carried out without the addition of a stabilizing agent at two different temperatures, 23 and 50 °C. The images taken under the optical microscope showed that the precipitated particles are metallic (Figure 1a). Additionally, a 3D photo of the obtained sediment was taken, which allowed to estimate the size of the precipitated particles from several dozen to several hundred microns (Figure 1b).

The effect of solution pH on the precipitation yields of PGM without stabilizing agent is shown in Figure 2. In an acidic solution (pH 3.5), the yields of Pd precipitation (Equation (1)) do not exceed 50%, however, as the pH increased to slightly acidic and then to alkaline, the amount of precipitated Pd increases. The increase in temperature significantly affected the precipitation only at pH 6.5, while the precipitation efficiency values at pH 3.5 and 8 are similar. The alkaline pH seems to have a more significant influence on the synthesis of NP than the temperature, which was also observed by other researchers [45].

The studies on Pd precipitation with HCOOH show that the metallic particles, larger than the nanosize, are obtained. Therefore, to obtain Pd-NP, the addition of a PVP stabilizing agent was proposed, as also suggested by other researchers [46,47,48]. The appropriate ratio of the stabilizing agent to the precursor may influence the morphology, structure, and particle size of the reduced material. For this purpose, the influence of the concentration of PVP on the precipitation yield was studied and is shown in Figure 3.

Regardless of the reducer used, in all cases the efficiency of Pd precipitationndecreases with the increasing concentration of PVP. In the case of Pt and Ru, the type of the reducer and the PVP concentration do not significantly affect the precipitation efficiency.

On the one hand, AA is an efficient reducer for platinum (P_Pt_ > 80%), on the other hand, AA is an inefficient reducer for palladium (P_Pd_ < 10%) (Figure 3a). This phenomenon can be helpful in the separation of platinum from palladium from the aqueous two-component solutions. The efficient reductor for Rh is NaBH_4_ and the precipitation efficiency of Rh is greater than 80%. The most pronounced change in the precipitation efficiency can be seen for Pd reduced with HCOONa. The reduction efficiency decreases as the molar ratio of PVP increases from 100 (for molar ratio of PGM:PVP:reducer was 1:1:2) to 20% (PGM:PVP:reducer was 1:5:2).

### 3.2. Effect of the Reducer on the Formation of PGM-NP

The influence of the type and concentration of a reducer on the formation of PGM-NP was investigated and the results for the precipitation of PGM from acidic solutions are presented in Table 1.

The concentration of the reducer has no significant influence on the precipitation yield for all PGMs. The most effective reducing agent for all metals is NaBH_4_ followed by HCOONa. However, due to the fact that AA shows low efficiency in the precipitation of Pd and Rh and has the potential to separate PGM by precipitation from their mixtures, it was chosen as a weak reducing agent for further research. For two selected reducers (AA and NaBH_4_) AFM analysis was used to determine the size of the obtained PGM particles. The results are presented in Table 2.

All obtained PGM-particles are nanoparticles; their size is below 90 nm. Pt-NPs after precipitation with NaBH_4_ are smaller than Pt-NPs after precipitation with AA. Of all samples, the smallest particles of 3 nm are obtained for Pd-NPs using AA or NaBH_4_ as reducing agent (molar ratio PGM:PVP:reducer 1:1:2). The largest particles for Ru are obtained after reduction with AA and are around 90 nm (molar ratio of PGM:PVP:reducer 1:1:2). Thus, the type of the reducer used affects in a significant way the size of the particles obtained. The reduction with NaBH_4_ and the PGM:PVP:reducer 1:1:2 molar ratio, 7–8 pH, ambient temperature are the best conditions for the formation of Pt, Pd, and Rh-NPs up to 5 nm. The change in the molar ratio of PVP to PGM from 2 to 5 does not result in a significant difference in the size of Rh NPs, while an increase in the NP size of Pt and Pd is observed. On the other hand, the conditions for the effective formation of the smallest Ru-NPs with a size of up to 4 nm cover the reduction with NaBH_4_ and the molar ratio of PGM:PVP:reducer 1:1:1, 7–8 pH, ambient temperature. Moreover, in this case, no significant influence of PVP ratio on the size of the obtained particles is observed. AFM images, confirming the particle size, are presented in the Appendix A (Figure A1).

TEM images are taken for some selected samples to check the structure and compare the particle size estimated with AFM technique (Figure 4).

TEM images of Pt or Pd-NP are used to determine the structure and size of the obtained particles. Particles precipitated with AA (Figure 4a,c) are spherical and have similar sizes of 7 nm. On the contrary, the particles precipitated with NaBH_4_ (Figure 4b,d) have a larger size range than these precipitated with AA, which are below 10 nm. In the case of Ru (Figure A2 in the Appendix A), the TEM images do not show clear structures of metal clusters as in the case of Pt, Pd and Rh (Figure 4). It can be seen that agglomerates of Rh-NP were also formed. Despite strong reducing agents, such as NaBH_4_, affecting the synthesis of smaller nanoparticles compared to weak reducing agents (e.g., AA), due to the high thermodynamic instability and excess surface energy, the synthesized particles can undergo nucleation and Ostwald ripening [49]. These phenomena lead to particle agglomeration, which can explain the larger agglomerates in samples after precipitation with NaBH_4_ than with AA. The trend of PGM particles to agglomerate has been observed by various researchers under various conditions [50,51,52,53]; it was found that agglomeration (for example of Pt-NPs) positively influenced the catalytic activity of the material in the CO oxidation reaction due to the high density of defects on the surfaces of the catalytic material [53].

Values of the particle size estimated from TEM images indicate that the NPs are much smaller than these estimated on the basis of AFM images. It is possible that the size obtained by AFM corresponded to aggregates in which single nanoparticles could not be observed. Due to a higher resolution of the TEM images than of the AFM ones, it is possible to distinguish between much smaller particles [54].

SEM-EDS analysis was used to confirm the presence of Pt in the metallic form of nanoparticles (Figure 5).

When the SEM-EDS images are compared with each other, it can be seen that oxygen and Pt do not overlap in the images. This indicates that the particles obtained do not form oxides but only metallic forms. In addition, the images show brighter points that can be attributed to the clusters of metals. Both TEM and SEM confirmed that the metal agglomerates formed during precipitation. A similar situation can be seen for Pd, Rh, and Ru-NPs, the presented images confirm that the obtained NPs are not metal oxides (Figure A3 in the Appendix A).

### 3.3. Two-Component Mixtures

The influence of the presence of two different PGMs in the stock solutions was investigated, and the results of the precipitation of both PGMs from acidic solutions are presented in Table 3.

Using precipitation with AA as a reducer, platinum, rhodium or ruthenium can be separated from palladium in two-component mixture. In all cases, the precipitation yield with NaBH_4_ is high for both PGMs in two-component mixture, therefore their separation is impossible. TEM images were taken for selected two-component samples (Figure 6) to compare the structure with single-component nanoparticles (Figure 4).

The materials shown in the TEM images have a structure similar to single-component nanoparticles (Figure 6). As in the previous images, the most agglomerated samples are obtained after reduction with NaBH_4_. TEM shows multilayer particle clusters (Figure 6a,b,d), as well as long agglomerate chains (Figure 6b). Only Pd/Rh-NPs after precipitation with AA does not form larger agglomerates. The particle size of the Pd/Rh-NPs synthesized in the two-component mixture differs in the range from 1 to 10 nm. However, the Pt/Rh-NPs size is centered more closely around 5 nm.

TEM images of Rh/Ru-NP after precipitation with AA and NaBH_4_ are presented in Figure A4 in the Appendix A. As in the case of a single-component solution with Ru-NP (Figure A2c,d), the TEM images do not show clear structures and metal clusters, although the Rh was agglomerating in a single mixture.

SEM-EDS images show rich, uniform precipitates of Pt-NPs in the sample (Figure 7a). The images also demonstrate bright structures from Rh particles (Figure 7a,b) which correspond to the metal clusters. On the other hand, Pd particles occur pointwise. By comparing the precipitation yield results and the SEM images, it can be confirmed that both Pt and Pd co-precipitated from Rh using NaBH_4_ as a reducing agent. Figure A5 in the Appendix A indicates the images of SEM-EDS Pt/Rh-NP and Pd/Rh-NP after precipitation with AA. Pt/Rh-NPs are similar to those precipitated with NaBH_4_ and both Pt and Rh rich precipitates are visible. Images of Pd/Rh-NPs reduced with AA show that Pd does not precipitate in a metallic form, but forms a likely soluble Pd compound, which is confirmed by small values of precipitation yield (Table 3).

### 3.4. Catalytic Properties of PGM-NPs 

An important characterization of the obtained PGM-NPs was the testing of their catalytic activity. An exemplary reduction reaction of NPh to APh was carried out in the presence of the synthesized nanoparticles. The course of the reaction was monitored by registration of UV-Vis spectra. The reference spectra of NPh and APh at pH 11.5 are compared to the spectra of the solutions during reaction after 1 or 30 min of reduction (Figure 8).

The maximum wavelength at 400 nm is attributed to NPh (in the sample before the reaction, pH 11.5) which shifted to 313 nm during the first minutes of reaction corresponding to an acidic form of NPh [42,44,55]. A maximum observed at 300 nm for the APh model APh solution is attributed to an acidic form of APh at 1.5 pH (Figure 8a).

A hypsochromic shift of the NPh maximum to lower wavelengths during the reaction was observed. The maximum at 260 nm corresponds to the basic (ionized) form of APh at 11.5 pH; it is clearly visible that, at the beginning of the reduction (after 1 min, Figure 8a), APh is not registered in the UV-Vis spectrum. However, the conversion of nitro groups to amino groups in the course of the reaction and, as a result, the significant decrease in NPh concentration decrease (decrease at maximum at 400 nm), are visible after 30 min of reduction (Figure 8b).

The degree of NPh conversion calculated according to Equation (3) proves that Pd-NPs provide the highest conversion degree of NPh conversion (about 90% after 5 min of reduction) compared to the NPs of other PGMs (Figure 9). The lowest conversion of NPh is reached when Ru particles are used. The order of the decreasing catalytic activity of the NPs is as follows: Pd > Rh~Pt >> Ru.

Research related to the separation of PGM nanoparticles from aqueous solutions by precipitation of PGM precipitation may be used in the future to develop a method to obtain PGM from solutions after leaching spent automotive catalysts. The perspective of recovering metals (especially PGM) from secondary resources is beneficial not only for the environment (less waste), but also for the economy (PGM recycling). It should be emphasized that the content of these valuable metals in waste (spent automotive converters) is several times higher than the content of PGM content in the richest available natural ores available in the world.

## 4. Conclusions

NaBH_4_ is an efficient reducer for all PGMs; the precipitation yields of Pt, Pd, Ru, Rh are 75, 90, 65, and 85%, respectively. The size of the PGM particles after precipitation with NaBH_4_ solution did not exceed 55 nm and the sizes of the PGM particles of PGM after precipitation with AA were below 90 nm. Precipitation of PGM with AA can be used for the separation of platinum from palladium from aqueous acidic solutions.

The development of the PGM nanoparticle precipitation method is very important for the development of knowledge in the field of separation processes, as it makes it possible to use aqueous solutions containing PGM (e.g., after leaching various waste materials-spent automotive converters) to produce catalytically active materials. The obtained Pt-NP, Pd-NP, and Rh-NP have the catalytic ability to catalyze the reaction of NPh to APh.

## Figures and Tables

**Figure 1 molecules-27-00390-f001:**
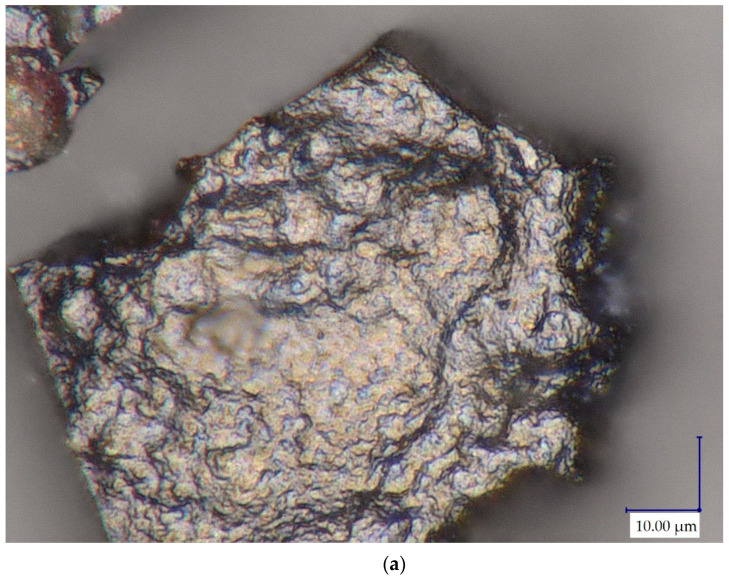
(**a**) Microscopic images (3000×) and (**b**) 3D image (200×) of a Pd precipitate reduced at Pd(II):HCOOH (1:2) ratio at 50 °C.

**Figure 2 molecules-27-00390-f002:**
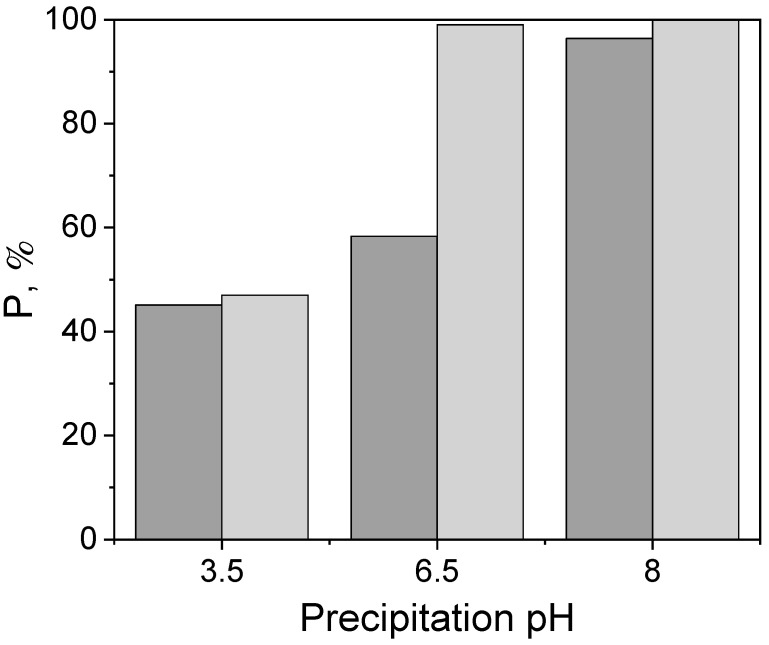
Influence of the pH of the solution on the efficiency of precipitation of Pd with 4 mM HCOOH at (■) 23 °C and (■) 50 °C.

**Figure 3 molecules-27-00390-f003:**
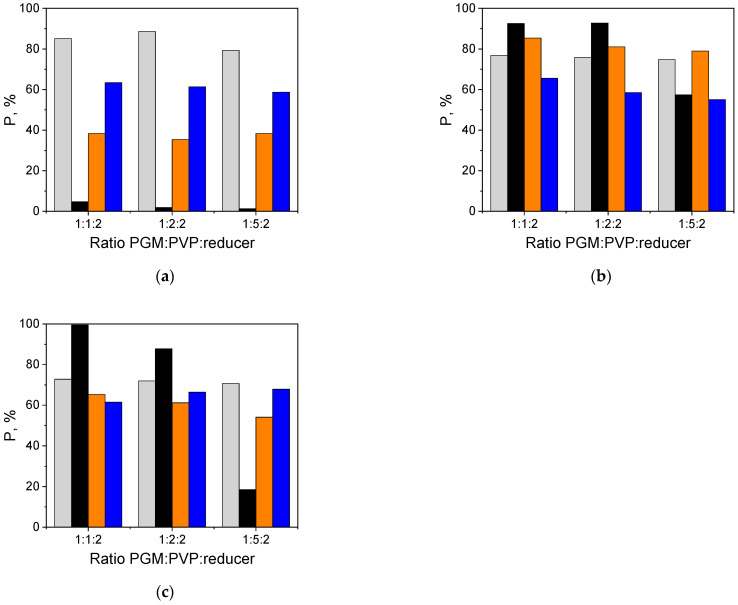
The effect of PVP concentration on the yield of (■) Pt-NP, (■) Pd-NP, (■) Rh-NP and (■) Ru-NP formation (the molar ratio of PGM:PVP:reducer was 1:1:2, 1:2:2 and 1:5:2, the reducer: (**a**) AA, (**b**) NaBH_4_, (**c**) HCOONa).

**Figure 4 molecules-27-00390-f004:**
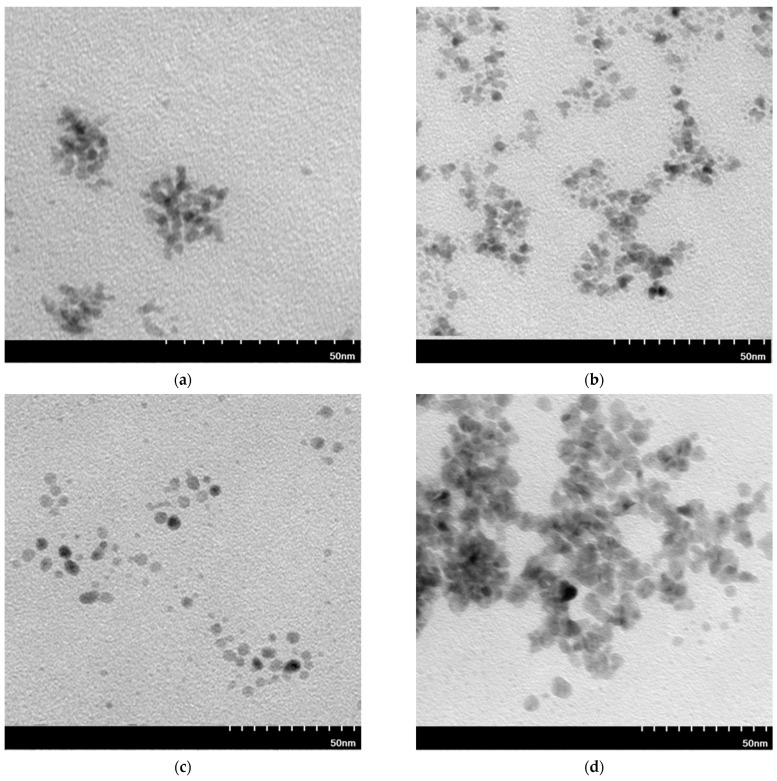
Transmission electron microscopy (TEM) images of Pt-NP (**a**,**b**) and Pd-NP (**c**,**d**) (the molar ratio of PGM:PVP:reducer was 1:5:2) using AA for (**a**,**c**) and (**c**) and NaBH_4_ for (**b**,**d**).

**Figure 5 molecules-27-00390-f005:**
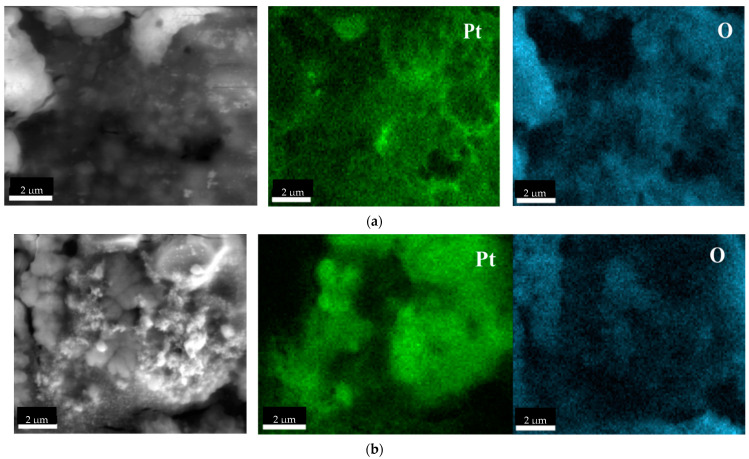
Scanning electron microscopy (SEM) coupled with EDS images of Pt-NP (the molar ratio of PGM:PVP:reducer was 1:5:2) using (**a**) AA and (**b**) NaBH_4_.

**Figure 6 molecules-27-00390-f006:**
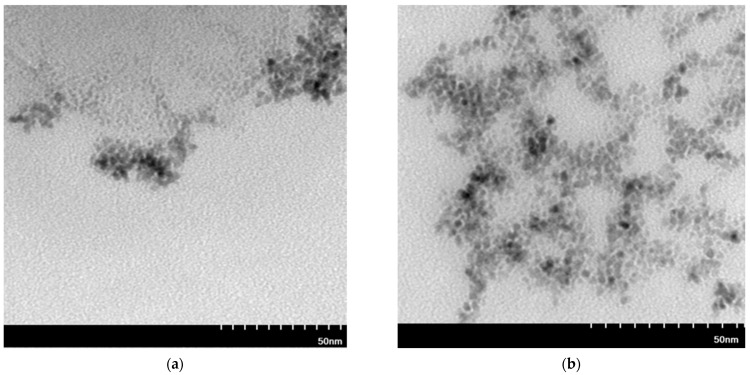
Transmission electron microscopy (TEM) images of Pt/Rh-NP (**a**,**b**) and Pd/Rh-NP (**c**,**d**) (the molar ratio of PGM:PVP:reducer was 1:1:1) using AA for (**a**,**c**) and NaBH_4_ for (**b**,**d**).

**Figure 7 molecules-27-00390-f007:**
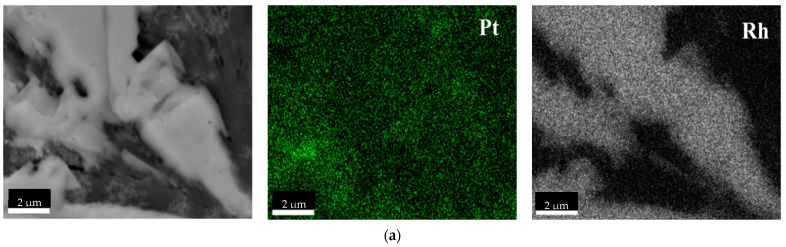
Scanning electron microscopy (SEM) coupled with EDS images of (**a**) Pt/Rh-NP and (**b**) Pd/Rh-NP (the molar ratio of PGM:PVP:reducer was 1:1:1) using NaBH_4_.

**Figure 8 molecules-27-00390-f008:**
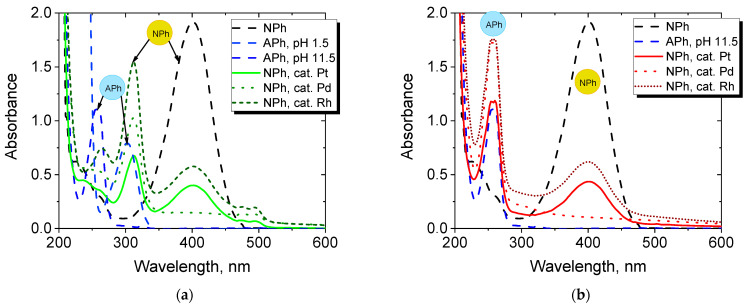
UV-Vis spectra of model solutions of NPh and APh in 11.5 pH and NPh solutions after (**a**) 1 min or (**b**) 30 min of reduction reaction catalyzed by synthesized Pt, Pd or Rh particles.

**Figure 9 molecules-27-00390-f009:**
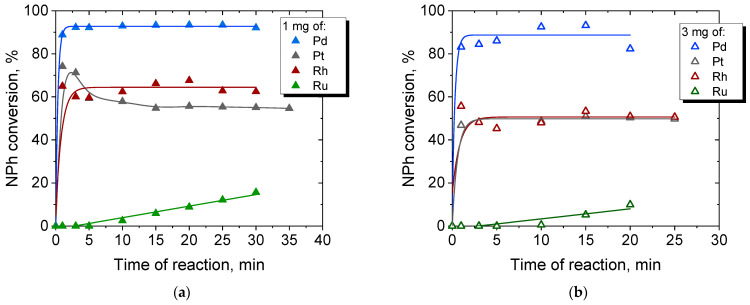
The conversion degree of NPh depending on the type of NPs used: (**a**) 1 mg of PGM and (**b**) 3 mg of PGM.

**Table 1 molecules-27-00390-t001:** Precipitation yield (P) of Pt, Pd, Rh, and Ru using one of three reducers: AA, NaBH_4_ and HCOONa.

Reducer	P, %			
	Pt	Pd	Rh	Ru
	Molar ratio of PGM:PVP:reducer 1:1:1			
AA	69.7	33.7	38.1	63.1
NaBH_4_	75.0	99.5	87.3	72.2
HCOONa	72.8	99.6	51.5	67.7
	Molar ratio of PGM:PVP:reducer 1:1:2			
AA	85.1	4.8	38.3	63.4
NaBH_4_	76.9	92.5	85.3	65.6
HCOONa	72.8	99.5	65.3	61.5

**Table 2 molecules-27-00390-t002:** Size of the precipitated particles obtained on the basis of the AFM (ambient temperature, pH 7–8).

Reducer	Particle Size, nm			
	Pt	Pd	Rh	Ru
	Molar ratio of PGM:PVP:reducer 1:1:1			
AA	39.5	10.7	16.5	33.5
NaBH_4_	8.1	20.9	40.5	3.6
	Molar ratio of PGM:PVP:reducer 1:1:2			
AA	46.9	3.1	52.1	88.3
NaBH_4_	3.2	3.2	5.2	53.3
	Molar ratio of PGM:PVP:reducer 1:5:2			
AA	40.3	14.9	23.9	39.3
NaBH_4_	7.3	36.3	5.4	43.7

**Table 3 molecules-27-00390-t003:** Precipitation yield of Pt, Pd, Rh and Ru from two-component mixture using one of two selected reducers: AA and NaBH_4_.

Reducer	P, %					
	Pt/Pd	Pt/Rh	Pt/Ru	Pd/Rh	Pd/Ru	Rh/Ru
AA	60.1/11.5	67.6/37.0	76.1/60.1	4.5/56.8	4.1/56.9	34.9/54.8
NaBH_4_	78.6/99.5	75.2/57.4	71.5/63.8	98.2/74.2	99.6/67.7	49.3/51.3

## Data Availability

The data presented in this study are available on request from the corresponding author.

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
