# Peer review of "Studies on the Formation of Catalytically Active PGM Nanoparticles from Model Solutions as a Basis for the Recycling of Spent Catalysts"

_molecules, 2022, doi:10.3390/molecules27020390_

Round 1

Reviewer 1 Report

The paper entitled ” Studies on the formation of catalytically active PGM nanoparticles from model solutions as a basis for the recycling of spent catalysts” treats the topic of precipitation of platinum group metals from acidic solutions. The obtained metallic particles are of nanometric sizes, thus can be more effectively used as catalysts for various chemical reactions then micrometric sized metallic particles.

The authors use model solutions for obtaining nanoparticles, which can serve as base for the recycling of spent automotive converters. Thus the topic is important and actual.

The Introduction cites about 40 papers, including the most recent ones. However my opinion is that it would serve better the understanding, if the introduction would be more focused on the presented topic, showing the advantages and disadvantages of the proposed method, showing the practical use of it, the possible applications of the proposed research, instead of only listing the papers found in the literature. The problem that the paper treats and tries to find an answer for, needs to be properly introduced. Therefore I suggest that the introduction would be changed accordingly.

The Results and discussion chapter treats the results, however instead of discussing the results of the actual experiments and measurements, a detailed comparison with the literature is done. Citing the literature should be part of the Introduction, the objective of the Results and discussion chapter is to find explanations, show what has been done, how and why.

In general the quality of the Figures is poor, especially the scale of the figures cannot be read, therefore the SEM and TEM imagen cannot prove what is stated in the text. Figure 1b, for example, is useless in its present form, therefore it doesn't show the sizes of the precipitates (lines 141, 142).

A clear discussion on the particles size needs to be done. What are the best circumstances for the formation of nanometric sized particles? Several unclear statements are done regarding the sizes. For example: in line 166 it is said that the size of the PGM particles was studied and it is shown in Figure 3, however Figure 3 shows the effect of the concentration on the yield.

AFM images could also be shown. Especially because the statements made in lines 206-209.

In the Table 2. only the particle sizes for two selected reducers are shown. Why?

HCOOH is listed as “used reducing agent” in 2.1, however the results obtained with it are not shown.

SEM-EDS has been used to build distribution maps of the metallic particles, and it is stated that the maps prove that the particles are present in metallic and not in oxide form. Why the authors did not show the SEM-EDS spectra for the metallic particles (or clusters)? The EDS spectra would be more convincing than the elemental distribution maps with not so ggod resolution.

The novelty of the article needs to be emphasized.

Author Response

Response to Reviewer 1 comments

The paper entitled ”Studies on the formation of catalytically active PGM nanoparticles from model solutions as a basis for the recycling of spent catalysts” treats the topic of precipitation of platinum group metals from acidic solutions. The obtained metallic particles are of nanometric sizes, thus can be more effectively used as catalysts for various chemical reactions then micrometric sized metallic particles.

The authors use model solutions for obtaining nanoparticles, which can serve as base for the recycling of spent automotive converters. Thus the topic is important and actual.

Question 1

The Introduction cites about 40 papers, including the most recent ones. However my opinion is that it would serve better the understanding, if the introduction would be more focused on the presented topic, showing the advantages and disadvantages of the proposed method, showing the practical use of it, the possible applications of the proposed research, instead of only listing the papers found in the literature. The problem that the paper treats and tries to find an answer for, needs to be properly introduced. Therefore I suggest that the introduction would be changed accordingly.

Author’s response: As advised by the Reviewer, we have added information about the innovation and the essence of the reduction method, as well as about some advantages and disadvantages of the chemical reduction method. Since we present fundamental research, the final application of the synthesized NPs has not been presented at the moment, however, the research confirms the catalytic properties of the newly synthesized materials. As the study compared both strong and weak reducers, in our opinion, it is important to introduce the readers to the reduction methods. Therefore, the paragraph related to the reduction methods was left in the Introduction.

Question 2

The Results and discussion chapter treats the results, however instead of discussing the results of the actual experiments and measurements, a detailed comparison with the literature is done. Citing the literature should be part of the Introduction, the objective of the Results and discussion chapter is to find explanations, show what has been done, how and why.

Author’s response: We thank the Reviewer for the suggestion. The way of referring to studies of other authors in Results and discussion has been changed. In Results and discussion we have left only the references that help us to explain our results and observations.

Question 3

In general the quality of the Figures is poor, especially the scale of the figures cannot be read, therefore the SEM and TEM images cannot prove what is stated in the text. Figure 1b, for example, is useless in its present form, therefore it doesn't show the sizes of the precipitates (lines 141, 142).

Author’s response: As advised by the Reviewer, we have improved the quality of the images. So, we hope, that now, the quality will be appropriate to get all the necessary pieces of information from the figures.

Question 4

A clear discussion on the particles size needs to be done. What are the best circumstances for the formation of nanometric sized particles? Several unclear statements are done regarding the sizes. For example: in line 166 it is said that the size of the PGM particles was studied and it is shown in Figure 3, however Figure 3 shows the effect of the concentration on the yield.

Author’s response: We thank the Reviewer for this helpful comment. We are sorry for the mistake. Of course, Figure 3 shows only precipitation yield dependence on the composition of the reducing solution. The results of the influence of PVP on particle size are presented in Table 2. So, the corrected sentence has been introduced instead of the previously stated in line 166, i.e. “For this purpose, the influence of the concentration of PVP on the precipitation efficiency and the size of PGM particles was studied and is shown in Figure 3 and Table 2, respectively.”. According to the results obtained in our studies, the best conditions to form nanometric PGM particles (Pt, Pd and Rh-NPs) are as follows: with NaBH4 and the PGM:PVP:reducer 1:1:2 molar ratio, 7-8 pH, ambient temperature. On the other hand, the conditions for the effective formation of the smallest Ru-NPs cover the reduction with NaBH4 and the molar ratio of PGM:PVP:reducer 1:1:1, 7-8 pH, ambient temperature. This information has been also added to the text of the manuscript.

Question 5

AFM images could also be shown. Especially because the statements made in lines 206-209.

            Author’s response: We thank the Reviewer for this advice. Because of it, we have added AFM images in Figure A1 in the Appendix, and the information about these images has been added to the manuscript.

Question 6

In the Table 2. only the particle sizes for two selected reducers are shown. Why?

Author’s response: We thank the Reviewer for this suggestion. Although, in the beginning, four reducers were selected for the studies, finally only two reducing agents, a weak one – ascorbic acid (AA) and a strong one – NaBH4, were chosen (on the basis of the precipitation yield and speed of reduction) for further research in order to compare two different reducing agents. This information has been added to the manuscript.

Additionally, AA seems to have the potential to separate PGM by precipitation from their mixtures because AA shows poor yield in the precipitation of Pd and Rh (up to 40%).

Question 7

HCOOH is listed as “used reducing agent” in 2.1, however the results obtained with it are not shown.

Author’s response: We are very sorry for the confusion that not all reducers were given. We added an information about HCOONa into Materials and Methods. We used four reducers: HCOOH, HCOONa, AA and NaBH4 in our research. For HCOOH and HCOONa, we presented only part of the research, because after preliminary research we focused mainly on precipitation with two selected reducing agents, i.e. AA and NaBH4. The results from HCOOH and HCOONa were added only to compare other reducing agents than AA and NaBH4.

Question 8

SEM-EDS has been used to build distribution maps of the metallic particles, and it is stated that the maps prove that the particles are present in metallic and not in oxide form. Why the authors did not show the SEM-EDS spectra for the metallic particles (or clusters)? The EDS spectra would be more convincing than the elemental distribution maps with not so good resolution.

Author’s response: We thank the Reviewer for this helpful comment. In our research, we wanted to prove that the obtained NPs are metallic, and not in the form of oxides. Unfortunately, we have only SEM-EDS images of the precipitated materials, and we do not have EDS spectra but we will take it into account in future research.

Question 9

The novelty of the article needs to be emphasized.

Author’s response: We thank the Reviewer for the helpful advice. Innovation and the impact of research on recycling have been added in the Introduction: “The development of an effective reduction method from leach solutions may have a significant impact on the recycling of precious metals from spent automotive converters and reuse of PGM, leading to limitation of metal mining from natural resources. Thus, the proposed fundamental research from various model solutions is vital to establish conditions which could be applied for the real leach solutions (e.g. after leaching of various waste materials - spent automotive converters) to produce catalytically active materials which may have a high application potential in the future, e.g. obtaining energy in the form of hydrogen through photoreforming from biomass or wastewater treatment from organic substances.”.

Also, please see the attachment. We added the manuscript with corrections (in red).

Reviewer 2 Report

  1. Page 2, line 64 : page 11 line 383 : need to be checked .
  2. Page 3, line 100: the full name AAS, AA in the first use.
  3. Although the subject of the manuscript is interesting. it needs to show the effect of other impurities in the catalyst leach liquor on the formation of nano particles .

Author Response

Response to Reviewer 2 comments

Question 1

Page 2, line 64: page 11 line 383: need to be checked.

Author’s response: As advised by the Reviewer, these corrections have been made in the manuscript.

Question 2

Page 3, line 100: the full name AAS, AA in the first use.

            Author’s response: We thank the Reviewer for this advice. It is corrected in the manuscript.

Question 3

Although the subject of the manuscript is interesting. It needs to show the effect of other impurities in the catalyst leach liquor on the formation of nano particles.

Author’s response: We thank the Reviewer for the suggestion. We are aware of the effect of the presence of other metal ions in the spent automotive catalyst on the NP formation. We have already started to conduct research related to the leaching of metal ions from spent automotive converters and the influence of non-precious metals on PGM precipitation from model and real leach solutions will be a subject of the further article. In the present manuscript, we focus mainly on fundamental research on precipitation of PGM from model solutions to set up the conditions for future studies on much complicated real solutions.

Also, please see the attachment. We added the manuscript with corrections (in red).

Round 2

Reviewer 1 Report

The Authors have taken into account the suggestions, answered the questions and improved the manuscript. The quality of the images has also been substantially improved. According to my opinion it can be accepted in the present form.